# Oxidative Stress-Induced Growth Inhibitor (OSGIN1), a Target of X-Box-Binding Protein 1, Protects Palmitic Acid-Induced Vascular Lipotoxicity through Maintaining Autophagy

**DOI:** 10.3390/biomedicines10050992

**Published:** 2022-04-25

**Authors:** Chong-Sun Khoi, Cai-Qin Xiao, Kuan-Yu Hung, Tzu-Yu Lin, Chih-Kang Chiang

**Affiliations:** 1Graduate Institute of Toxicology, College of Medicine, National Taiwan University, Taipei 106, Taiwan; d05447005@ntu.edu.tw (C.-S.K.); r05447013@ntu.edu.tw (C.-Q.X.); 2Department of Anesthesiology, Far-Eastern Memorial Hospital, New Taipei City 22060, Taiwan; drling1971@gmail.com; 3Department of Internal Medicine, College of Medicine and Hospital, National Taiwan University, Taipei 106, Taiwan; kyhung@ntu.edu.tw; 4Department of Integrated Diagnostics & Therapeutics, National Taiwan University Hospital, Taipei 10002, Taiwan

**Keywords:** palmitic acid, OSGIN1, unfolded protein response, XBP1s, autophagy, eNOS, endothelial cells

## Abstract

Saturated free fatty acids (FFAs) strongly correlate with metabolic syndromes and are well-known risk factors for cardiovascular diseases (CVDs). The mechanism of palmitic acid (PA)-induced vascular lipotoxicity under endoplasmic reticulum (ER) stress is unknown. In the present paper, we investigate the roles of spliced form of X-box-binding protein 1 (XBP1s) target gene oxidative stress-induced growth inhibitor 1 (OSGIN1) in PA-induced vascular dysfunction. PA inhibited the tube formation assay of primary human umbilical vein endothelial cells (HUVECs). Simultaneously, PA treatment induced the XBP1s expression in HUVECs. Attenuate the induction of XBP1s by silencing the XBP1s retarded cell migration and diminished endothelial nitric oxide synthase (eNOS) expression. OSGIN1 is a target gene of XBP1s under PA treatment. The silencing of OSGIN1 inhibits cell migration by decreasing phospho-eNOS expression. PA activated autophagy in endothelial cells, inhibiting autophagy by 3-methyladenine (3-MA) decreased endothelial cell migration. Silencing XBP1s and OSGIN1 would reduce the induction of LC3 II; therefore, OSGIN1 could maintain autophagy to preserve endothelial cell migration. In conclusion, PA treatment induced ER stress and activated the inositol-requiring enzyme 1 alpha–spliced XBP1 (IRE1α–XBP1s) pathway. OSGIN1, a target gene of XBP1s, could protect endothelial cells from vascular lipotoxicity by regulating autophagy.

## 1. Introduction

Intracellular and extracellular lipid homeostasis are major determinants of cellular health. Calorie overload without proportional calorie utilization results in the increased storage of fatty acids in adipocytes [1]. Excess fatty acids accompanied by triglyceride accumulation in parenchymal cells of multiple tissues, including skeletal myocytes, cardiac myocytes, hepatocytes, and pancreatic beta cells, results in chronic cellular dysfunction and injury, a process known as lipotoxicity [2]. In addition to causing damage to various organs, saturated free fatty acids, such as palmitic acid (PA), can induce endothelial dysfunction through different mechanisms, such as PA-induced reactive oxygen species (ROS) generation via the calcium 2+/protein kinase C-alpha/NADPH oxidase 4 (Ca^2+^/PKCα/NOX4) pathway, to impair endothelial function [3]. PA also induces endothelial cell apoptosis by impairing autophagy flux [4]. Vascular endothelial dysfunction plays a major role in atherosclerosis progression [5]. In addition, the activation of endothelial cells by hypercholesterolemia also contributes to the development of atherosclerosis [6].

Inflammation, insulin resistance, and endoplasmic reticulum stress (ER stress) contribute to lipotoxicity [7,8]. Numerous environmental, physiological, and pathological insults disrupt ER homeostasis, known as ER stress. The unfolded protein response (UPR) is a cellular stress response related to ER stress [9]. Protein homeostasis or proteostasis, which maintains cellular function and structure, is regulated via the UPR pathway in the ER [10]. Inositol-requiring enzyme 1α (IRE1α) is one of three arms of the UPR signaling pathway that activates and IRE1α possesses endoribonuclease spliced XBP1 mRNA, which follows the translation of spliced XBP1 (XBP1s). XBP1 plays a vital role in maintaining proteostasis, such as XBP1 suppresses the expression of pathological tau via endoplasmic reticulum-associated degradation (ERAD), to maintain proteostasis [11]. Neuronal XBP1s increases lysosomal acidity in the intestine to improve proteostasis [12]. In the vascular system, XBP1s plays multiple roles in endothelial cell proliferation, autophagy response, and apoptosis [13]. XBP1s inhibits tumor necrosis factor alpha (TNFα)-induced inflammation and leukocyte adhesion in human retinal endothelial cells [14]. Additionally, miR-214 suppresses angiogenesis and endothelial function by downregulating the expression of XBP1s [15]. Additionally, XBP1 interacts with histone deacetylase 3 (HDAC3) to protect endothelial cells from disturbed flow-induced oxidative stress [13]. Furthermore, Toru et al. found that the incorporation of XBP1s into exosomes is transmitted extracellularly when cells are exposed to stress [16]. However, the sustained activation of XBP1s leads to endothelial apoptosis and atherosclerosis development [17].

Oxidative stress-induced growth inhibitor 1 (OSGIN1), known as ovary kidney and liver protein 38 (OKL38), is highly expressed in the ovary, kidney, and liver [18]. OSGIN1 is a downstream gene of p53, which could elevate cellular ROS production; meanwhile, OSGIN1 interacts with p53 to facilitate cytochrome c release from mitochondria leading to apoptosis under stress [19]. In endothelial cells, the upregulation of OSGIN1 by oxidized phospholipid increased superoxide production, [20] and oxidized fatty acid epoxyisoprostane E2 (EI)-induced oxidative stress is nuclear factor erythroid 2-related factor 2 (Nrf2) dependent [21]. However, OSGIN1 could protect endothelial cells against oxidative stress after oxidized phospholipid treatment [22], but its molecular mechanism role remains unclear during free fatty acid-induced vascular lipotoxicity.

Although ER stress has been reported in lipotoxicity [23,24,25,26], the role of XBP1 and its target gene OSGIN1 in vascular lipotoxicity remains vague. In the present study, we show that treatment with PA results in XBP1 splicing and enhances the expression of OSGIN1. XBP1s-regulated OSGIN1 protects endothelial cells from lipotoxicity by regulating autophagy.

## 2. Materials and Methods

### 2.1. Cell Culture

Primary human umbilical vein endothelial cells (HUVECs) were obtained from the Food Industry Research and Development Institute (Hsinchu, Taiwan). HUVECs were cultured with M199 medium containing 10% fetal bovine serum (FBS, Biological Industries, Boca Raton, FL, USA) on 1% gelatin-coated (Sigma-Aldrich, St. Louis, MO, USA) plates. Meanwhile, 20 μg/mL endothelial cell growth supplement (ECGS, Merck Millipore, Massachusetts, USA), 50 μg/mL heparin (Sigma-Aldrich, St. Louis, MO, USA), and 1% penicillin/streptomycin (Gibco, New York, NY, USA) were added to the culture medium. The cells were cultured in an incubator at 37 °C, humidified, containing 95% air and 5% CO2. The cultured cells were harvested for the following experiments at passages 6 to 8.

### 2.2. Bovine Serum Albumin (BSA)-Conjugated Palmitate Preparation

Sodium palmitate (Sigma-Aldrich, St. Louis, MO, USA) was conjugated with bovine serum albumin (BSA, essentially fatty acid free; Sigma-Aldrich, St. Louis, MO, USA) to make a water-soluble fatty acid complex. First, BSA was added to 10 mL of 150 mM NaCl solution, and then this solution was stirred at a temperature of 37 °C. The sodium palmitate was added to 5 mL of 150 mM NaCl solution and clarified to 70 °C. This palmitate solution was added to another 5 mL BSA solution and continued to be stirred at a temperature of 37 °C for 1 h. Finally, the solution pH was adjusted to 7.4, and the final BSA-conjugated PA complex was stored at −20 °C before use in the subsequent experiments.

### 2.3. MTS Assay

The quantification of cell proliferation was performed by using the 3- (4,5-dimethylthiazol-2-yl)-5- (3-carboxymethoxyphenyl)-2- (4-sulfophenyl)-2 H-tetrazolium, inner salt (MTS) assay. The reduction in the MTS tetrazolium dye by NAD(P)H-dependent dehydrogenase enzymes generated a colored formazan product in living cells. Ten thousand HUVECs were plated in each well of the 96-well plate. Fresh culture medium containing 0.2% MTS (Promega, WI, USA) was added to each well and incubated for 30 min at 37 °C. The SpectraMax^®^ 190 microplate reader was used to measure absorbance at 490 nm and 650 nm.

### 2.4. Oil Red O Staining

Cellular lipid deposition was assessed by oil red O staining. First, the cultured cells were fixed with 4% paraformaldehyde for 30 min. After removing the paraformaldehyde, isopropanol was added and incubated for 5 min. After the removal of isopropanol, we then covered the cells with oil red O solution (Sigma-Aldrich, St. Louis, MO, USA) for 5 min at room temperature. We discarded the oil red O solution, then added the hematoxylin for 1 min. This was washed with PBS and we observed the culture cell under the microscope.

### 2.5. Wound Healing Migration Assay

HUVECs were seeded in gelatin-coated 12-well plates and cultured until confluence. The monolayer was gently and slowly scratched in a straight line with a new 1 mL pipette tip, followed by removing the debris and smoothing the edge of the scratch. The cells were then replaced with a fresh conditioned medium containing BSA or PA and incubated at 37 °C for 24 h. The images were captured from three different sites of each scratching line at 0 and 24 h. The gap distance was quantitatively evaluated using Image J. The average migrated distance was calculated on three sites/line × three lines/well × three wells.

### 2.6. Tube Formation Assay

According to the manufacturer’s instructions, the Matrigel matrix (Corning) was melted overnight at 4 °C. Then, the Matrigel solution was added to a 96-well plate with 50 μL/well and solidified at 37 °C for 30 min. The cell suspension containing a total of 3000 HUVECs was added to Matrigel-coated wells in triplicate for 24 h in a 37 °C incubator. Light microscope (Leica, Wetzlar, Germany) was used to evaluate tube formation and photographed by using the LAS v4.6 Image software (Leica, Wetzlar, Germany); the tube numbers were counted and quantified by ImageJ.

### 2.7. Small Interfering RNA (siRNA) Transfection

To prepare siRNA complexes, 60 nM of siXBP1 or siNTC was mixed with 100 μL of transferred buffer, and the T20 transfection reagent (Alisa, Bioscience, Taipei, Taiwan) was subsequently added to siXBP1 or siNTC. The mixture was incubated for 15 min at room temperature. The cell culture medium was removed, and serum-free DMEM (2.5 mL) and 200 µL of siRNA complexes were added to each well. After transfection for 3 h, the medium was removed, and M199 medium with 10% FBS was added and incubated for 48 h. Finally, HUVECs were treated with BSA or PA for 24 h for subsequent experiments.

### 2.8. Quantitative Real-Time Polymerase Chain Reaction (qRT-PCR)

Total RNA was extracted from cultured cells using GENEzol reagent and TriRNA Pure Kit (Geneaid, New Taipei City, Taiwan) following the manufacturer’s protocol. According to the manufacturer’s instructions, cDNA was reverse transcribed from RNA using a reverse transcriptase kit (iScript cDNA synthesis kit, BIO-RAD, Hercules, CA, USA). The thermal cycling reaction was set as follows: 10 min at 95 °C, followed by 40 cycles at 95 °C for 15 s and 60 °C for 1 min. The synthesized cDNA was stored at −20 °C and was later used as a template. TaqMan gene expression assay (Life Technologies, Carlsbad, CA, USA) was used to perform RT-PCR. cDNA was used as templates, and the thermal cycling reaction of StepOne Plus System (Life Technologies, Carlsbad, CA, USA) was set for 2 min at 50 °C, followed by 10 min at 95 °C, 36 cycles of 15 s at 95 °C, and 1 min at 60 °C. The following PCR primers were used in this study: XBP1 forward: 5’ TGCTGAGTCCGCAGCAGGTG 3’ and XBP1 reverse: 5’- GGGGCTTGGTATATATGTGG; OSGIN1 forward: GACCTGGAGGTCAAGGACTG, OSGIN1 reverse: CACTCCACGGCTGTGACTAC; ANGPTL4 forward: TCCGTACCCTTCTCCACTTG, and ANGPTL4 reverse: CCAGGTCTTCCAGAAGATTCC; TXNIP forward: GCAAGCCTAATGGCTACTCG, TXNIP reverse: CGGTCAAGAAAAGCCTTCAC; CCND2 forward: GTGCTGTCTGCATGTTCCTG, CCND2 reverse: CTGCCAGGTTCCACTTCAAC; LC3B forward: AGCAGCATCCAACCAAAATC, LC3B reverse: CTGTGTCCGTTCACCAACAG; and 3’GAPDH forward: 5-GGCATGGACTGTGGTCATGA-3 and GAPDH reverse: 5-TGACTCCACTCACGGCAAAT-3′.

### 2.9. Western Blotting Analysis

HUVEC whole cells were lysed with radioimmunoprecipitation assay (RIPA) buffer (20 Mm Tris-HCl, 150 mM NaCl, 1 Mm EDTA, 0.1% SDS, 1% NP40, 1% sodium deoxycholate) containing protease inhibitors (Roche, Basel, Switzerland). The samples were incubated at 4 °C for 10 min and then centrifuged at 14,000 rpm for 10 min at 4 °C. The supernatant was removed and followed by washing with PBS. Bradford assay (Thermo Fisher Scientific, Portsmouth, NH, USA) was used to quantify the protein concentrations. The protein in lysate samples was separated by using sodium dodecyl sulfate-polyacrylamide gel electrophoresis (SDS-PAGE) and transferred to a methanol-activated polyvinyl difluoride (PVDF) membrane (0.45 μm) (Merk Millipore, Burlington, MA, USA). The following membrane was blocked with 5% milk in TBST (20 mM Tris-HCl, 150 Mm NaCl, 0.05% Tween, pH 7.4) at room temperature, then the blots were incubated with anti-eIF2α (Santa Cruz, 1:2000); anti-phospho-eIF2α (Cell Signaling, 1:2000); anti-CHOP (Cell Signaling, 1:2000); anti-XBP1 (Abcam, 1:2000); anti-OSGIN1 (Genetex, 1:1000); anti-phospho-eNOS (Cell Signaling, 1:2000); anti-LC3 (1:1000), anti-Beclin-1 (1:1000); anti-ATG5 (1:1000); anti-ATF6 (Novus Biologicals, 1:2000); anti-GAPDH (GeneTex, 1:10,000); and anti-α-tubulin (Sigma Aldrich, 1:10,000), followed by incubation with the HRP-conjugated secondary antibody. Finally, the membrane was scanned with UVP BioSpectrum 810 (DBA Analytik, Jena, Upland, CA, USA) and quantified by Image J.

### 2.10. Immunohistochemistry (IHC) Staining

Human artery tissue microarrays were purchased from Biomax (AR301), including normal artery and atherosclerosis tissue samples. First, the samples were heated, followed by deparaffinized with xylene and rehydrated using alcohol. Citrate buffer (pH 6.0) was used for antigen retrieval for heating 15 min and cooled down. A total of 3% H_2_O_2_ was used to remove peroxidase and blocked with 3% BSA in PBS. The sample was incubated with the primary antibody of OSGIN1 (Thermo Fisher Scientific, 1:100) at 4 °C overnight. After washing with PBS, horseradish peroxidase (HRP) was used to conjugate the OSGIN1 antibody and counterstained with hematoxylin. These staining slides were scanned with TissueFAXS (TissueGnostics, Los Angeles, CA, USA) and quantified by Image J.

### 2.11. Intracellular Reactive Oxygen Species (ROS) Assay

Dichlorodihydrofluorescein diacetate (DCFDA) is a fluorescent compound used to detect cellular reactive oxygen species (ROS) generation. HUVECs (5 × 10^3^) were cultured in a 96-well culture dish for 24 h and transfection with siRNA of OSGIN1 for another 48 h. Subsequently, 20 μM of DCFDA was added for 30 min, then washed twice with PBS to remove the DCFDA containing medium and replaced with no phenol red medium. Next, 0.25 μM of PA was added and cultured for 24 h. Finally, the fluorescence intensity was measured at 485 nm/535 nm (excitation/emission spectra) by using a Paradigm Multi-Mode Plate Reader.

### 2.12. Statistical Analysis

The individual values of related figures were shown in Appendix A. The two-tailed Student’s *t*-test was used to compare two groups. One-way ANOVA, followed by Tukey’s post hoc test, was used to compare more than two groups. All statistical analyses were performed using GraphPad Prism version 8 (Intuitive Software for Science, San Diego, CA, USA). All data are presented as mean ± S.E.M. * *p* < 0.05 was considered statistically significant.

## 3. Results

### 3.1. PA Activates ER Stress-Related UPR Signaling in Endothelial Cells

Accordingly, PA can induce ER stress in different cell types [23,24,25]. To investigate whether PA triggers ER stress in HUVECs, several ER stress-related UPR markers, such as eukaryotic translation initiation factor 2α (eIF2α), XBP1, and activating transcription factor 6 (ATF6), were analyzed using Western blotting. HUVECs were treated with 0.125–0.5 mM of PA for 24 h. BSA was used as a negative control. The activation of protein kinase R (PKR)-like endoplasmic reticulum kinase (PERK) phosphorylated α subunit of eukaryotic initiation factor 2 (eIF2α) when UPR occurred [27]. The results show that the protein expression of phosphorylated eIF2α is enhanced in PA-stimulated cells in a dose-dependent manner, compared to that in BSA-treated cells (Figure 1A). In response to ER stress, a portion of the membrane-bound 90 kDa ATF6 undergoes proteolytic cleavage, releasing the soluble 50 kDa protein with an ability to induce transcription [28]. The p50 ATF6 protein expression was markedly increased following PA treatment in a dose-dependent manner (Figure 1B). Similar to PERK, inositol-requiring 1 α (IRE1α) is one of the ER transmembrane receptors during UPR, which activates the IRE1α spliced mRNA of XBP1 through endoribonuclease, and XBP1 eventually translates to active proteins [27]. The result show the active spliced form of XBP1 was induced by PA treatment in a dose-dependent manner (Figure 1C). We also investigated the role of the C/EBP homologous protein (CHOP), which regulated ER stress-associated apoptosis during the severe impairment of ER functions [29]. The cells treated with PA (0.5 mM) induced the transcription factor CHOP (Figure 1D). These results collectively demonstrate that the UPR pathway of ER stress can be successfully activated by PA treatment in HUVECs.

### 3.2. Treatment with PA Results in Endothelial Dysfunction

In a previous study, PA has been shown to induce necroptosis in endothelial cells [30]. The MTS assay was used to examine the cytotoxicity of PA on HUVECs. The results show that cell viability is significantly reduced following treatment with 0.5 mM PA, but not in 0.25 mM PA (Figure 2A). Additionally, 0.25 mM of PA increased intracellular lipid accumulation (staining by oil red O) after 24 h (Figure 2B). Hence, 0.25mM PA was the optimal dose used in the following study. We further investigated the migration of endothelial cells by using the wound-healing assay. The results show a slower migration of PA-treated cells than BSA-treated control cells (Figure 2C). In addition, the angiogenesis of endothelial cells was evaluated by tube formation assay. PA attenuated tube formation in HUVECs, as shown by the formation of fewer tubular structures (Figure 2D).

### 3.3. IRE1α–XBP1s Axis Activation Protects Endothelial Dysfunction Following PA Exposure

The chronic activation of ER stress in endothelial cells leads to increased oxidative stress and inflammation, often resulting in cell death [31]. In contrast to the detrimental effects of ER stress, the protective effect of ER stress via the activation of XBP1s preconditioning could protect against retinal endothelial inflammation [14]; hence, XBP1s was selected for further experiments. To determine whether the IRE1α–XBP1s axis is involved in PA-induced endothelial dysfunction, XBP1s was silenced via siRNA transfection to elucidate the role of XBP1s in endothelial function. Transfection with siRNA effectively suppressed PA-induced XBP1 expression in mRNA (Figure 3A) and protein levels (Figure 3B). Meanwhile, the silence of XBP1s resulted in a significant reduction in cell migration (Figure 3C). eNOS mediated cell migration, which contributes to angiogenesis [32]; in the present study, Western blot analysis was used to verify the role of eNOS in cell migration under the silencing of XBP1s during PA-induced vascular lipotoxicity. The silencing of XBP1s inhibited the expression of eNOS and p-eNOS, with eNOS expression specifically decreasing remarkably (Figure 3D). These results prove that XBP1s protect PA-induced endothelial dysfunction through eNOS expression by evidence of silenced XBP1s.

### 3.4. XBP1s Protects Endothelial Cells from Lipotoxicity via Autophagy

PA induces Ca^2+^-dependent autophagy, which results in programed necrosis (necroptosis) in endothelial cells [30]. However, Zhou et al. demonstrated that resveratrol attenuates endothelial oxidative injury by inducing autophagy [33], and ER stress-induced autophagy is associated with cytoprotection [34]. LC3 II and p62 are autophagy markers, of which LC3 II participates in the membrane formation of autophagosomes and p62 binds with ubiquitinated protein during autophagy [35]. In addition, beclin-1 is one of the components of class III PI(3)K complexes that are involved in the maturation of autophagosome [36] and autophagy-related protein 5 (ATG5) conjugated autophagy-related protein 12–autophagy-related protein 16 (ATG12–ATG16) complex promotes autophagosome formation [37]. Then, we investigated the role of PA-induced autophagy further by using Western blot analysis through these autophagy-related markers. The results show that the expression of LC3 II increases after PA exposure, which decreases after co-treatment with 4-methyl umbelliferone 8-carbaldehyde (4u8c), which is a selective IRE1α inhibitor (Figure 4A). We then investigated whether XBP1s triggered the autophagic signaling pathway under lipotoxic conditions through the silencing of XBP1s. The expressions of LC3 II and beclin-1 decreased after the silencing of XBP1s (Figure 4B,C). The expression of ATG 5 decreased after the silencing of XBP1s, but this was statistically insignificant (Figure 4D). The role of autophagy in endothelial cells was evaluated using the wound healing assay. Autophagy was inhibited by 3-methyladenine (3-MA) (Figure 4E), and the inhibition of autophagy decreased cell migration after PA exposure (Figure 4F). Collectively, these results indicate that XBP1s protects endothelial cells via autophagy under lipotoxic conditions through the silencing of XBP1s.

### 3.5. XBP1s Regulated OSGIN1 against Endothelial Dysfunction via Autophagy

To search for the XBP1s-regulated molecular signaling in response to PA, RNA-seq analysis was used to find the target genes of XBP1s by the silencing of XBP1s after PA treatment. The XBP1s target genes were selected at the threshold > 2.0 log2 fold change in the condition of silencing XBP1s, compared to normal cells. The result show that 19 genes meet the criteria described in the heat map with log2 fold change (Figure 5A) and the Differentially Expressed Gene (DEGs) in BSA vs. PA-treated group was shown in Appendix A. We further validated four interesting candidate target genes: angiopoietin-like 4, OSGIN1, cyclin D2, and thioredoxin-interacting protein (ANGPTL4, OSGIN1, CCND2, and TXNIP). Among these potential genes, OSGIN1 was the most remarkably increased in mRNA level (Figure 5B). From a previous study, oxidized phospholipid enhanced the expression of OSGIN1 in the human aortic endothelial cells [20], but the molecular mechanism of OSGIN1 remains unclear under free fatty acid-induced vascular lipotoxicity. We also found that PA promoted OSGIN1 expression and decreased the expression of OSGIN1 after the silencing of XBP1s in Western blot analysis (Figure 5C). This result determines that XBP1s regulates OSGIN1. The silencing of OSGIN1 via siRNA transfection was explored to elucidate its role and molecular function in vascular endothelial cells. The result shows that OSGIN1 could be silenced effectively by siRNA at the mRNA level. The LC3 B mRNA level increased after silencing OSGIN1, but there was no statistical significance (Figure 5D). Then, we found the decreased protein expression of p-eNOS and LC3 II after silencing OSGIN1 (Figure 5E). Meanwhile, the silencing of OSGIN1 suppressed cell migration under BSA and PA treatment (Figure 5F). Collectively, these results indicate that the PA-enhanced expression of OSGIN1 was regulated by XBP1s, which obviously modulate cell migration through autophagy. Additionally, OSGIN1 could regulate endothelial cell migration via p-eNOS expression by evidence of silenced OSGIN1.

PA induced ROS generation through NADPH oxidase 2 (NOX2) and NADPH oxidase 4 (NOX4) [38], but the role of OSGIN1 in regulated ROS production remains vague under vascular lipotoxicity; hence, we examined the intracellular ROS under the silencing of OSGIN1 after PA treatment by using the DCF-DA assay. The result shows that PA-induced ROS production was indicated by increased fluorescence intensity after 24 h exposure. However, ROS generation was not significantly higher or lower under the silencing of OSGIN1 with PA treatment (Figure 5G). This result determines that OSGIN1 is not involved in ROS generation in endothelial cells.

### 3.6. OSGIN1 Is Involved in the Atherosclerosis Process

Hyperlipidemia acts as a major risk factor of atherosclerosis [39]; furthermore, it also leads to vascular endothelial dysfunction [40], which is involved in the development and progression of atherosclerosis [5,6]. Our present study demonstrated that OSGIN1 is upregulated during free fatty acid-induced endothelial dysfunction, but whether OSGIN1 participated in atherosclerosis is unknown. Hence, we analyzed OSGIN1 expression in artery tissue by using tissue microarrays. We found that the atherosclerosis artery had a higher expression of OSGIN1 compared to the normal artery, especially in the endothelial layer (Figure 6A). This result suggests that OSGIN1 participates in the atherosclerosis process, which may protect or preserve the endothelial function.

To clarify the regulation between molecular interaction, Ingenuity Pathway Analysis (IPA) was used for pathway analysis. The pathway analysis demonstrated that PA activates XBP1s, OSGIN1, and autophagy. Meanwhile, PA inhibits the expression of nitric oxide synthase 3 (NOS3). XBP1s direct-regulated LC3B may modulate autophagy under PA treatment (Figure 6B). These analysis results are consistent with our present study. Additionally, we firstly found that OSGIN1 was regulated by XBP1s, which could affect autophagy and phospho-eNOS expression under PA treatment. However, these intermolecular pathways were not shown in the IPA pathway analysis. In addition, OSGIN1 did not participate in PA-induced ROS generation.

## 4. Discussion

Cardiovascular diseases (CVDs) are a major factor affecting mortality worldwide. Endothelial dysfunction is an early event in the progression of multiple CVDs. Saturated free fatty acids (FFAs) are highly correlated with metabolic syndromes and are well-known risk factors of CVDs [5]. The endothelium is among the first targets of elevated FFA levels in the blood that mediate endothelial cell dysfunction and death. FFAs mediate endothelial cell death and dysfunction through various molecular mechanisms [41,42,43]. Therefore, targeting the pathogenic signaling pathways involved in FFA-induced endothelial dysfunction might be a preventive strategy for protection against endothelial dysfunction and endothelial dysfunction-related cardiovascular complications. Our study demonstrated that PA could activate XBP1s and promote OSGIN1 expression in endothelial cells. Silencing XBP1s and OSGIN1s decreased endothelial cell migration, and XBP1s-dependent OSGIN1 maintains endothelial cell function by regulating autophagy (Figure 7).

Autophagy is thought to be a pro-survival stress response that helps cells maintain intracellular metabolic homeostasis under stress conditions [44]. From a previous study, endothelial autophagic flux under high shear stress limits atherosclerotic plaque formation by preventing endothelial apoptosis, senescence, and inflammation [45]. In addition, autophagy serves as a cytoprotective mechanism against the endothelial-to-mesenchymal transition (EndMT) to promote angiogenesis by reducing the expression of Snail under hypoxic conditions in human cardiac microvascular endothelial cells [46]. However, in certain circumstances, autophagy induction may also promote endothelial cell death, such as PA-activated Ca^2+^-dependent autophagy leading to the necroptosis of endothelial cells with the depletion of ATP [30]. Additionally, PA-induced lipotoxicity is associated with autophagy activation, which enhances ROS generation leading to the inhibition of nitric oxide (NO) production, cell migration, and tube formation [3], and, also, PA-induced senescence of endothelial cells through disturbed autophagic flux by the inhibited fusion of autophagosomes and lysosomes [47]. Furthermore, Zhao et al. demonstrated that PA could activate autophagy but suppressed c-Jun amino-terminal kinases (JNK), p-38 dependent autophagic flux, leading to cell apoptosis [4]. In contrast, Song et al. found that PA inhibits the AMP-activated protein kinase/mammalian target of rapamycin (AMPK/mTOR)-regulated autophagy, which subsequently promoted intracellular ROS production and decreased the expression of p-eNOS. Resveratrol restored autophagy through activated AMPK to attenuate ROS production and enhanced p-eNOS, which improved endothelial dysfunction caused by PA [33]. Similar to this previous study, our present study demonstrates that PA-induced autophagy may protect endothelial cells from vascular lipotoxicity by evidence of the inhibition of autophagy-reduced endothelial migration (Figure 4F).

The literature reviews implicate that XBP1s has a protective role in vascular homeostasis and disease [48,49]. Additionally, the latest study reveals that XBP1s could activate autophagy through Transcription factor EB (TFEB), which may contribute to glucose tolerance and steatosis in in vivo and in vitro studies [50]. During Japanese encephalitis virus infection in the neuronal cells, activated autophagy by XBP1s and ATF6 could attenuate virus-induced cell death [51]. In cancer therapy, disulfiram form complex with copper (DSF/Cu) induce the apoptosis of cancer cell lines through activated IRE1–XBP1s regulated autophagy [52]. In addition, XBP1s induces autophagy in endothelial cells by directly binding to the BECLIN-1 gene promoter [53]. These previous studies were consistent with our result showing that the silencing of XBP1 decreases the expression of beclin-1 (Figure 4C) and attenuates cell migration (Figure 3C). Taken together, our result shows a similar protective role of XBP1s-regulated autophagy against PA-induced vascular lipotoxicity. However, Zeng et al. demonstrated that XBP1 overexpression downregulated VE-cadherin expression and induced apoptosis in endothelial cells [17]. Moreover, the activation of XBP1s leads to parallel increases in the expression of the monocyte chemoattractant protein 1 (MCP-1) via histone methylation, which is related to the pathogenesis of diabetic nephropathy [54]. These findings indicate a two-pronged influence of XBP1s on endothelial cells. Further investigations are required for elucidating this critical point and a new pharmacological strategies involving endothelial cells.

OSGIN1 is a tumor suppressor that induces apoptosis through translocation to mitochondria and increasing the release of cytochrome c; meanwhile, OSGIN1 is associated with hepatocellular carcinoma (HCC) progression and patient survival [55]. Our results firstly reveal that OSGIN1 is a target of XBP1s (Figure 5C), but intermolecular regulation could be proved by future studies. From a previous study, OSGIN1 protects endothelial cells from oxidized phospholipid-induced oxidative stress [22]; our result also demonstrates that OSGIN1 counteracts vascular lipotoxicity to maintain cell migration (Figure 5F). In addition, OSGIN1 also regulates the autophagy process by enhancing autophagosome formation in ROS-dependent pathways, which are associated with the activation of the AMPK/mTOR signaling pathway after docosahexaenoic acid (DHA) treatment in breast cancer cells [56]. Additionally, OSGIN1 promoted apoptosis, cell cycle arrest, and the autophagy of human bronchial epithelial (HBE) cells and alveolar basal epithelial cells (A549) after particulate matter (PM) 2.5 treatment, which was regulated by methyltransferase-like 3 (METTL3) [57]. OSGIN1 was regulated by long non-coding RNA (lnRNA) UCA1 to mitigate autophagy flux-mediated apoptosis through the mTOR pathway in human hepatocellular carcinoma cells (HepG2) under arsenic toxicity [58]. To determine the role and participation of OSGIN1 in vascular autophagy, we also reported that, for the first time, the silencing of OSGIN1 inhibited cell migration by regulating autophagy (Figure 5E,F) under PA treatment. Numerous studies have verified PA-induced ROS generation and oxidative stress [38,59,60,61]. In addition, OSGIN1 overexpression contributed to elevating cellular ROS in human bone osteosarcoma epithelial cells (U2OS cells) [19]. Intriguingly, our study demonstrated that OSGIN1 did not contribute to ROS generation (Figure 5G). From these literature reviews and our present study, we implicated that OSGIN1 possess the ability to fight against cardiovascular disease through preserving endothelial cell function.

Based on the previous reports, eNOS plays a critical role in vascular endothelial function. ROS-mediated eNOS uncoupling contributes to the development of cardiovascular disease; thus, several studies are trying to develop therapeutic strategies by targeting eNOS [62]. The activation of the phosphoinositide 3-kinases/serine/threonine-specific protein kinases/endothelial nitric oxide synthase (PI3K/Akt/eNOS) signaling pathway by hydrogen sulfide protects endothelial cells from high glucose-induced ROS generation, ER stress, apoptosis, and inflammatory responses [63]. Duan et al. also proved that the activation of PI3K/Akt/eNOS by andrographolide mitigated high glucose-disturbed cell migration, apoptosis, and inflammation in endothelial cells [64]. Our study demonstrates that XBP1s preserved the endothelial cell function through eNOS expression (Figure 3D). Meanwhile, we reported that, for the first time, silencing OSGIN1 remarkably decreased p-eNOS protein levels under PA treatment, suggesting a regulatory role of OSGIN1 in mediating p-eNOS expression. Our results disclose that OSGIN1 is a target of XBP1s, but the silencing of XBP1s is not able to downregulate p-eNOS expression; these results may implicate that the XBP1s/OSGIN1 axis modulates endothelial cell migration, mainly through autophagy, but not p-eNOS. Further studies could be conducted in the future to investigate intermolecular signaling.

Autophagy can regulate and affect the expression of eNOS, such as endothelial shear stress-induced autophagy, maintain the phosphorylation of eNOS and bioavailability of NO, while it can also suppress ROS production and inflammatory cytokines in a study in vitro [65]. Additionally, steady laminar shear stress upregulated autophagy to modulate the expression of eNOS and decrease endothelin 1 (ET-1) in an ex vivo study [66]. However, Liu et al. disclosed that autophagy induced by angiotensin II in HUVEC suppresses NO production; this autophagy process may contribute to endothelial dysfunction by inhibiting p-eNOS [67]. In contrast, eNOS also regulates autophagy, such as ischemic postconditioning enhancing AMPK/eNOS-mediated autophagy to mitigate the apoptosis of cardiac cells and decrease the infarct size/area during ischemic/reperfusion injury [68]. Autophagy and eNOS would affect each other under different circumstances; however, our present study shows that both are regulated by the upstream role of OSGIN1 to maintain endothelial function during vascular lipotoxicity.

Endothelial dysfunction is the first step of atherosclerosis; meanwhile, the endothelium is the earliest target during atherosclerosis formation [69]. Accumulated evidence showed that atherosclerosis is a major cause of CVD disease [70]; our immunohistochemistry result firstly revealed a high expression of OSGIN1 in the human atherosclerosis artery (Figure 6A), which implicated that OSGIN1 may preserve or maintain endothelial function during the atherosclerosis process.

## 5. Conclusions

The accumulation of PA lead to XBP1 splicing and enhanced OSGIN1 expression in endothelial cells. XBP1s regulated OSGIN1 and significantly downregulated autophagy, which influenced the angiogenesis of endothelial cells. In the present study, we demonstrated a new mechanism by which XBP1s-modulated OSGIN1 plays a protective role in maintaining endothelial cell function under lipotoxic conditions. We highlighted the important the XBP1s/OSGIN1 axis signaling pathway as a promising therapy to prevent endothelial dysfunction-related CVDs during vascular lipotoxicity.

## Figures and Tables

**Figure 1 biomedicines-10-00992-f001:**
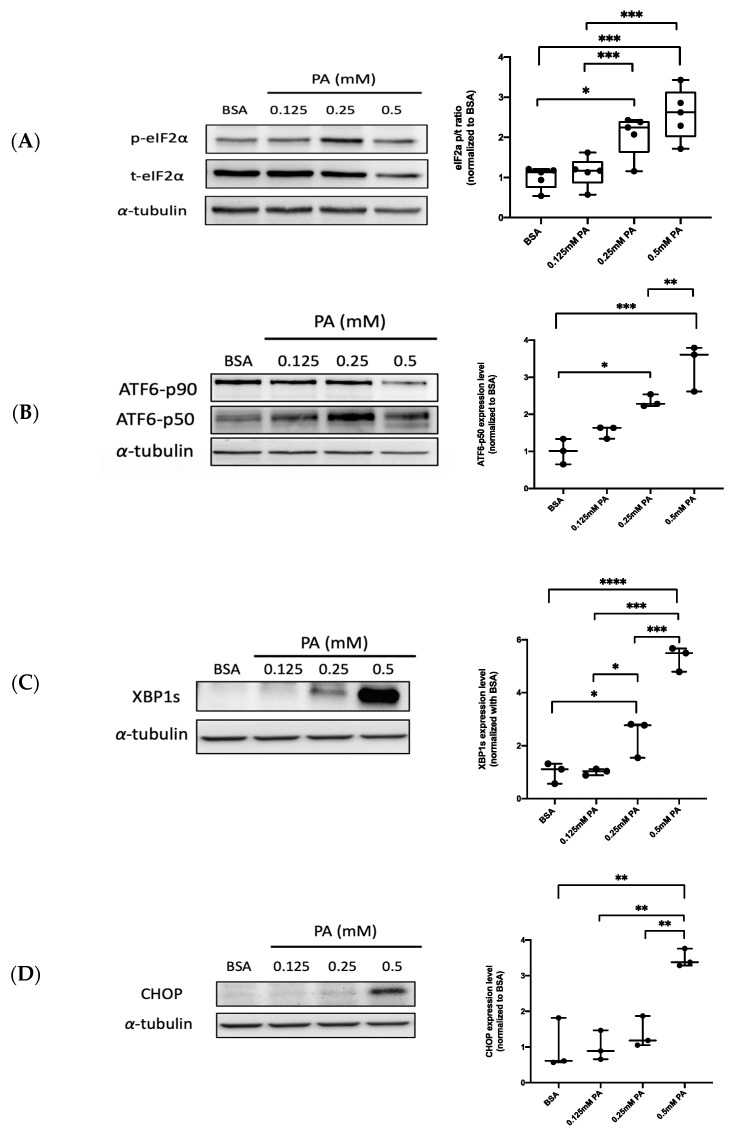
PA activates three pathways of unfolding protein response (UPR) signaling in HUVECs. PA was administered to HUVECs in different concentrations (0.125–0.5 mM) for 24 h. The protein expression of ER stress-related UPR markers, including (**A**) phosphorylated and total eIF2α, (**B**) ATF6, (**C**) spliced XBP1, and (**D**) CHOP, was analyzed using Western blotting and quantified. Data are presented as the mean ± SEM obtained from three independent groups. Data from multiple groups were analyzed by one-way ANOVA, followed by Tukey’s post hoc test. * *p* < 0.05, ** *p* < 0.01, *** *p* < 0.001, **** *p* < 0.0001.

**Figure 2 biomedicines-10-00992-f002:**
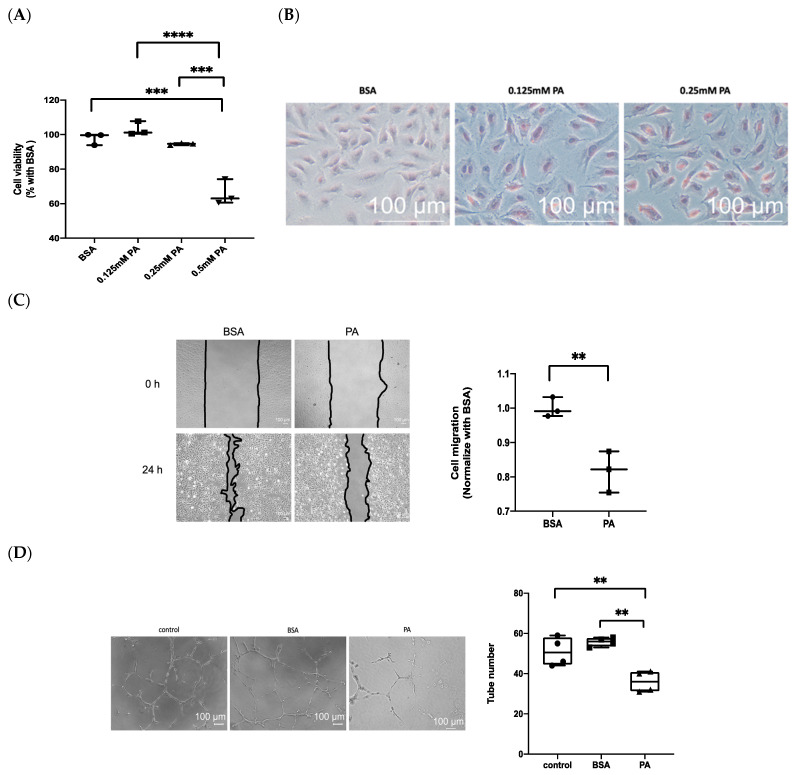
Palmitic acid (PA) treatment induces endothelial dysfunction in HUVECs. (**A**) The cytotoxic effects of PA on HUVECs were examined using the MTS assay. The HUVECs were treated with PA (0.125–0.5 mM) for 24 h; BSA was used as the control. (**B**) Intracellular lipid deposition was observed using oil red O (orange-red, scale bar: 100 μm). (**C**) Wound healing assay was performed in the presence of BSA or PA and quantification. (**D**) The tube formation of endothelial cells (scale bar: 100 μm) and quantification. Data are presented as the mean ± S.E.M of three independent experiments. Data from two groups were analyzed using the two-tailed Student’s *t*-test. Data from multiple groups were analyzed by one-way ANOVA, followed by Tukey’s multiple post hoc test. ** *p* < 0.01, *** *p* < 0.001, **** *p* < 0.0001.

**Figure 3 biomedicines-10-00992-f003:**
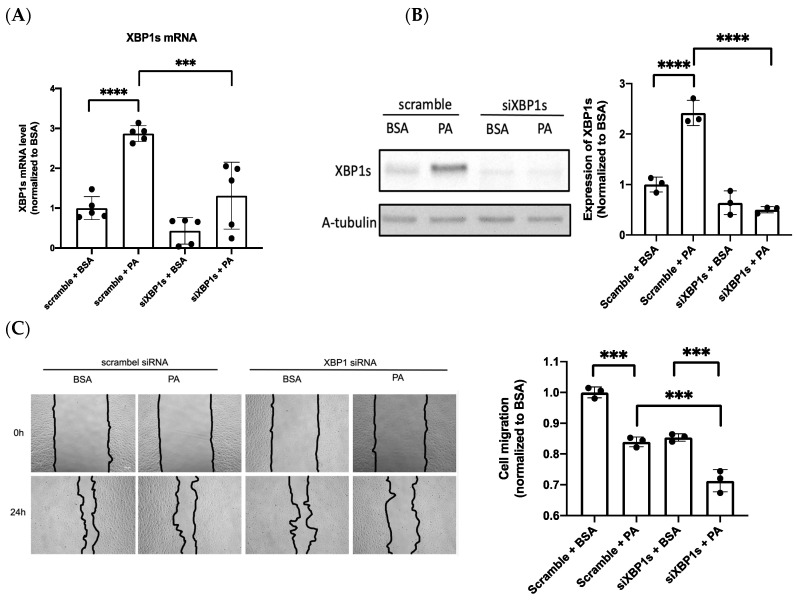
The IRE1α–XBP1 axis protects endothelial dysfunction in PA-exposed cells. The silencing of XBP1s by siRNA was analyzed using (**A**) qRT-PCR and (**B**) Western blotting. (**C**) The wound healing assay was observed at 0 h and 24 h post-scratching after silencing XBP1s under PA treatment (scale bar: 100 μm). (**D**) Protein expression of eNOS, p-eNOS after the silencing of XBP1s. Data are presented as the mean ± SEM, and the images are representative of three independent groups. Data from multiple groups were analyzed by one-way ANOVA, followed by Tukey’s post hoc test. * *p* < 0.05, *** *p* < 0.001, **** *p* < 0.0001, respectively. ns: not significance.

**Figure 4 biomedicines-10-00992-f004:**
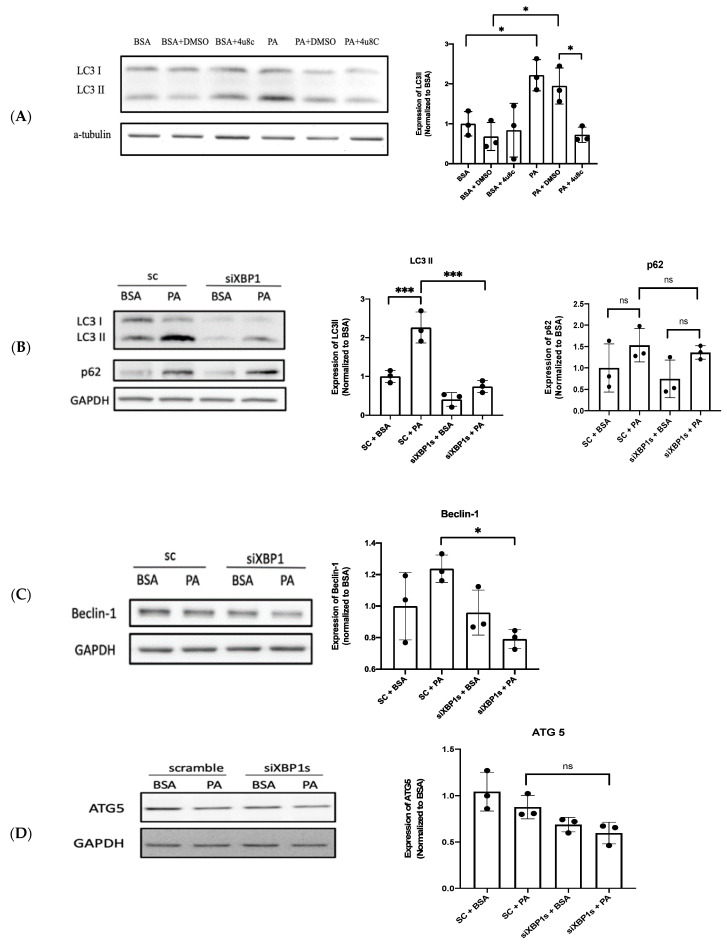
XBP1 protects endothelial cells from lipotoxicity by regulating autophagy. Autophagy induced by PA was analyzed using Western blotting and quantified. The protein expression of (**A**) LC3 II/I after the co-treatment of PA and 4u8c; (**B**) LC3 II/I, p62 after the silencing of XBP1s, (**C**) beclin-1, (**D**) ATG5, and (**E**) LC3 II/I after co-treatment with PA and 3-MA; and (**F**) wound healing assay after combined treatment with PA and 3-MA. Data are presented as the mean ± SEM, and the images are representative of three independent groups. Data from multiple groups were analyzed by one-way ANOVA, followed by Tukey’s post hoc test. * *p* < 0.05, *** *p* < 0.001, **** *p* < 0.0001, respectively. ns: not significance.

**Figure 5 biomedicines-10-00992-f005:**
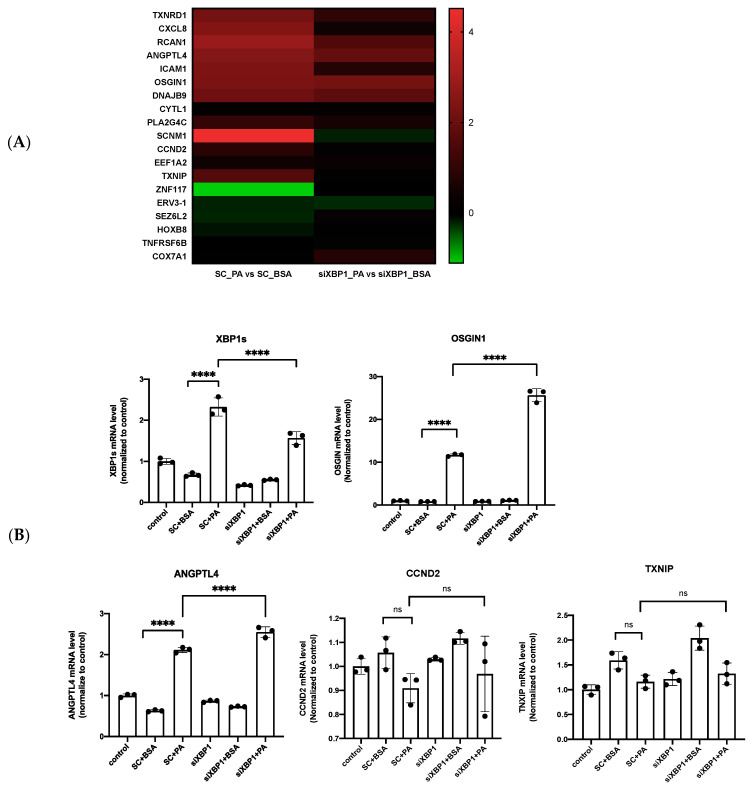
PA-induced OSGIN1 against endothelial dysfunction. (**A**) The heatmap demonstrates 19 XBP1s target genes whose expression > 2.0 log2 fold changes in the silencing of XBP1s (siXBP1s), compared to siRNA control cells after PA treatment. The mRNA levels of (**B**) XBP1s, ANGPTL4, CCND2, TXNIP, (**B**,**D**) OSGIN1, and (**D**) LC3B were analyzed with qRT-PCR. (**C**) The protein expression of OSGIN1 after the silencing of XBP1s. (**E**) The protein expression of OSGIN1, p-eNOS, and LC3 II/I after the silencing of OSGIN1. (**F**) Wound healing assay after silencing OSGIN1 combined treatment with PA. (**G**) Intracellular ROS was detected by DCFDA assay after silencing OSGIN1 combined treatment with PA. Data are presented as the mean ± SEM, and the images are representative of three independent groups. Data from multiple groups were analyzed by one-way ANOVA, followed by Tukey’s post hoc test. * *p* < 0.05, ** *p* < 0.01, *** *p* < 0.001, **** *p* < 0.0001, respectively. ns: not significance.

**Figure 6 biomedicines-10-00992-f006:**
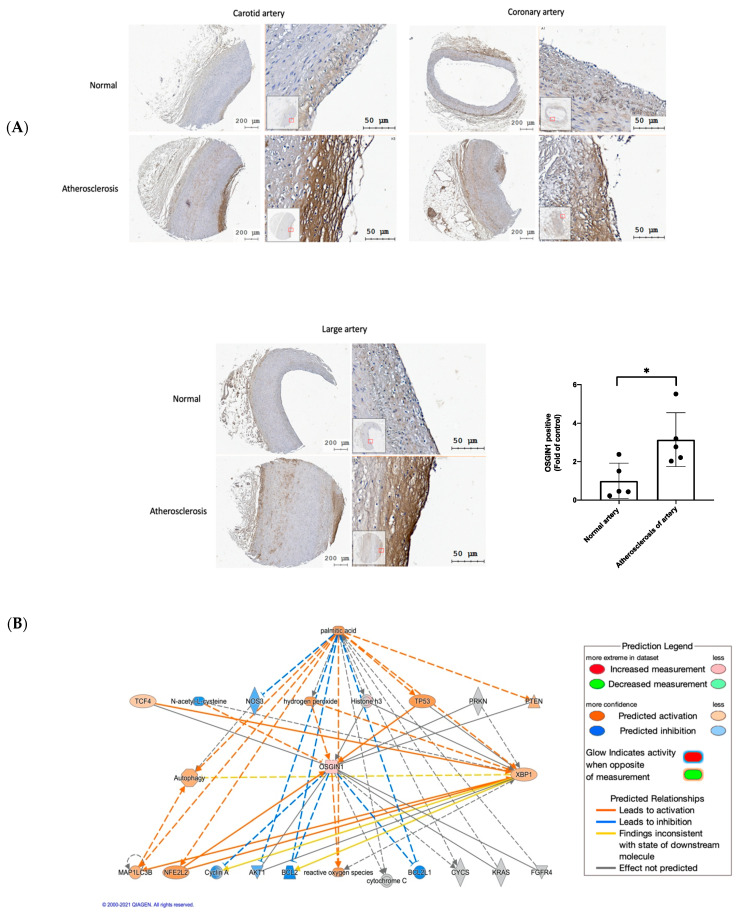
OSGIN1 is involved in atherosclerosis processing: (**A**) The expression of OSGIN1 in human artery tissue microarrays was determined with immunohistochemistry (normal artery *n* = 5, atherosclerosis artery *n* = 5, respectively; scale bar: 50 µm). (**B**) QIAGEN’s Ingenuity Pathway Analysis was used to generate the pathway analysis. Data from two groups were analyzed with two-tailed Student’s *t*-test. * *p* < 0.05 compared to control.

**Figure 7 biomedicines-10-00992-f007:**
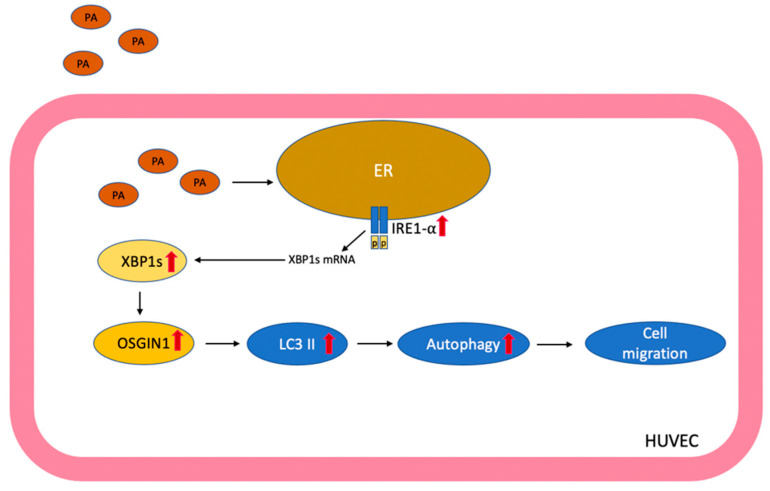
Schematic diagram depicting that the mechanisms for XBP1s regulates OSGIN1 and protects endothelial cells from PA-induced lipotoxicity. The accumulation of PA initiated UPR in endothelial cells; meanwhile, PA activated the IRE1α–XBP1 axis, leading to alternative XBP1 mRNA splicing, and the upregulation of the XBP1s protein. PA enhanced the expression of OSGIN1, which was regulated by XBP1s to maintain cell migration through autophagy under PA-induced vascular lipotoxicity.

## Data Availability

Not applicable.

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
