# Peer review of "Oxidative Stress-Induced Growth Inhibitor (OSGIN1), a Target of X-Box-Binding Protein 1, Protects Palmitic Acid-Induced Vascular Lipotoxicity through Maintaining Autophagy"

_biomedicines, 2022, doi:10.3390/biomedicines10050992_

Round 1

Reviewer 1 Report

The authors have explained about the involvement of OSGIN1 in PA-induced vascular lipotoxicity. Aim of this article is interesting, but there are several critical concerns that the authors should address as follows.

Comment 1

I think the explanations in Figure 1B and Figure 1C are reversed. Please confirm this point.

Comment 2

XBP1 and IRE1 are the proteins of ER stress sensors. I think that PA induces ER stress in this study because PA activates IRE1-XBP1 pathway. However, in this article, the authors don’t describe the reason that PA induces ER stress. Please explain the point in detail based on your current knowledge in this article.

Comment 3

In figure 3D and 5E, the addition of PA is thought to cause phosphorylation of eNOS, but the results in this article show no change or decrease. Moreover, although there was no change in p-eNOS using XBP1 siRNA, p-eNOS expression was decreased by using the siRNA of OSGIN1, which is downstream of XBP1. The authors should explain the reason for this result.

Comment 4

In figure 3A and 5B, the expression level of XBP1 mRNA is increased by using PA, although the silencing is performed using XBP1 siRNA. I think that RNAi suppresses the expression of the corresponding mRNA. Please explain the reason for the increase in this result.

Comment 5

In figure 5B, if OSGIN1 is located downstream of XBP1, RNAi results suggest that silencing of XBP1 results in similar or lower OSGIN1 mRNA levels compared to SC. Consider why this result is increasing.

Comment 6

In figure 5D and E, the amount of LC3 protein decreased even though the amount of LC3 mRNA did not change. Please explain the reason for this point.

Comment 7

In figure 5F, when SC + PA and siOSGIN1 + PA are compared, there is a significant difference in cell migration. However, it is difficult to understand that it is essentially changing both visually and numerically, and it remains a question whether it can be interpreted that migration is suppressed. Please show any other data related to the migration of OSGIN1.

Comment 8

Please check how to write the unit. For example: 4 °C → 4°C (cannot open space), 15s → 15 s (open space)

Comment 9

Please put a space before the parentheses.

Reviewer 2 Report

BIOMEDICINES ARTICLE REVIEW

Title: OSGIN1, a downstream target of XBP1s, protects PA-induced vascular lipotoxicity through maintaining autophagy and eNOS expression

Authors: Khoi et al.

The work presented by Khoi et al. attempts to demonstrate the role of the XBP1s/OSGIN1 axis in the protection of palmitic acid-induced vascular lipotoxicity through observations on autophagy and endothelial nitric oxide synthase expression.

To do this, the authors first analyse the parameters affected by palmitic acid-induced endoplasmic reticulum stress, finding significant increases in the expression levels of translation initiation factor 2α (eIF2α), phosphorylated and total transcription factor 6 (ATF6), transcription factor CHOP and the spliced form of X-box binding protein 1.

Secondly, they analyse the effects of palmitic acid on cultures of primary human umbilical vein endothelial cells (HUVEC). Here, they find a low cytotoxicity effect as the IC50 for palmitic acid is around 0.5 mM, as a consequence, the authors choose the concentration of 0.25 mM, with a cytotoxicity of almost o% for the rest of the studies. At the same time, they analyse intracellular lipid accumulation, endothelial cell migration, as well as angiogenesis.

Subsequently, they silence XBP1 leading to a significant reduction in cell migration and eNOS expression. In addition, XBP1 silencing causes decreases in LC3II expression increases caused by palmitic acid, although PA expression is still increased after XBP1 silencing. Finally, they show increases in OSGIN1 levels that are greater in XBP1-silenced cells in the presence of PA.

Throughout the manuscript there are many contradictions that the authors should make a great effort to explain and correct if necessary.

Formal aspects of the manuscript

1.- Title: The title should always reflect the main conclusion of the work and use as few acronyms as possible, especially those that are not in the public domain and could confuse readers. Authors should therefore modify the title significantly. As a sample, I suggest the following: "Oxidative stress-induced growth inhibitor (OSGIN1), a target of the X-box binding protein 1, protects palmitic acid-induced vascular lipotoxicity by maintaining autophagy and eNOS expression".

2.- In this regard, the authors overuse acronyms (is it necessary to abbreviate palmitic acid in the title? PA can be many things) as if any reader were up to date with them. As a general rule, authors should reduce the number of acronyms to a minimum and, above all, define all of them first and then use them both in abbreviated and extended form and alternatively in order not to make the reader lose the idea.

3.- In some cases they do not even define the abbreviations, which makes it very difficult to understand what the authors want to communicate. Examples, 3MA, 4u8C... etc.

4.- A very important thing is that authors should always explain why they use certain molecular or genetic parameters and what their meaning is. And I say this for practically all those used.

5.- As long as the authors do not correct these points from this formal point of view, I will not be able to approve the acceptance of the manuscript.

6.- The authors abuse the word "downstream" and it is not at all necessary in this context, therefore, it must be corrected.

Background aspects of the work

1.- It is very difficult to understand that the negative effects of a compound, in this case palmitic acid, on vascular lipotoxicity is also the solution. If PA causes increases in XBP1 levels, and this lipotoxicity is then characterised by increases in the IRE1-α/OSGIN1 axis, it cannot be said that this axis protects against lipotoxicity. In any case, it should be said that the blockage of this axis is responsible for the protection. And this should be made explicit both in the title and throughout the text.

2.- It is very strange that with only three experiments, on the one hand, such a large statistical range is obtained in most cases (the authors should show the individual values of these experiments in supplementary files), and on the other hand, that in some cases there are significant differences.

3.-An important example of this occurs in an important case for the work, i.e., variations in eNOS expression. The authors say that there are very significant differences when it does not seem so. In particular, figure 3D. It is very difficult to believe that with the expression data shown there could be significant differences between siXBP1+BSA versus aiXBP1 +PA. Reanalyse similar ones.

4.- Throughout the manuscript there are sentences that we do not understand why they are included, or at least the authors do not clarify or justify them adequately. For example, in Results 3.3 there is a sentence that is not adequately explained: "eNOS-mediated cell migration contributes to angiogenesis [29]".

5.- The schematic figure should be modified because, if PA causes increases in ATF6, IRE1-α, elF-2α, XBP1a, OSGIN1 and LC3II, these markers cannot, at the same time, be the protectors of the lipotoxicity caused by PA itself.

Author Response

Response to reviewers’ comments:

Dear Editor-in-chief:

Many thanks for the hard work and numerous useful suggestions you provided on our manuscript. We followed the reviewers’ suggestions, and the detailed corrections are listed below, point-by-point, based on the issues have been raised.

Response to Reviewer 2’s Comments:

The work presented by Khoi et al. attempts to demonstrate the role of the XBP1s/OSGIN1 axis in the protection of palmitic acid-induced vascular lipotoxicity through observations on autophagy and endothelial nitric oxide synthase expression.

To do this, the authors first analyse the parameters affected by palmitic acid-induced endoplasmic reticulum stress, finding significant increases in the expression levels of translation initiation factor 2α (eIF2α), phosphorylated and total transcription factor 6 (ATF6), transcription factor CHOP and the spliced form of X-box binding protein 1.

Secondly, they analyse the effects of palmitic acid on cultures of primary human umbilical vein endothelial cells (HUVEC). Here, they find a low cytotoxicity effect as the IC50 for palmitic acid is around 0.5 mM, as a consequence, the authors choose the concentration of 0.25 mM, with a cytotoxicity of almost o% for the rest of the studies. At the same time, they analyse intracellular lipid accumulation, endothelial cell migration, as well as angiogenesis.

Subsequently, they silence XBP1 leading to a significant reduction in cell migration and eNOS expression. In addition, XBP1 silencing causes decreases in LC3II expression increases caused by palmitic acid, although PA expression is still increased after XBP1 silencing. Finally, they show increases in OSGIN1 levels that are greater in XBP1-silenced cells in the presence of PA.

Throughout the manuscript there are many contradictions that the authors should make a great effort to explain and correct if necessary.

Formal aspects of the manuscript

1.- Title: The title should always reflect the main conclusion of the work and use as few acronyms as possible, especially those that are not in the public domain and could confuse readers. Authors should therefore modify the title significantly. As a sample, I suggest the following: "Oxidative stress-induced growth inhibitor (OSGIN1), a target of the X-box binding protein 1, protects palmitic acid-induced vascular lipotoxicity by maintaining autophagy and eNOS expression".

Ans: We rewrote the title as “Oxidative stress-induced growth inhibitor (OSGIN1), a target of the X-box binding protein 1, protects palmitic acid-induced vascular lipotoxicity by maintaining autophagy and eNOS expression (line 2- line 5)" which suggested by reviewer.

2.- In this regard, the authors overuse acronyms (is it necessary to abbreviate palmitic acid in the title? PA can be many things) as if any reader were up to date with them. As a general rule, authors should reduce the number of acronyms to a minimum and, above all, define all of them first and then use them both in abbreviated and extended form and alternatively in order not to make the reader lose the idea.

Ans: We rechecked and rewrote the full names before using the abbreviation.

3.- In some cases they do not even define the abbreviations, which makes it very difficult to understand what the authors want to communicate. Examples, 3MA, 4u8C... etc.

Ans: We define the abbreviations like 3-methyladenine (3-MA) (line 27, line 411), “4-methyl umbelliferone 8-carbaldehyde (4u8C) which is selective IRE1αinhibitor (line 405-line 406)”.

4.- A very important thing is that authors should always explain why they use certain molecular or genetic parameters and what their meaning is. And I say this for practically all those used.

Ans: We added the sentences as “activation of protein kinase R-like endoplasmic reticulum kinase (PERK) phosphorylated α subunit of eukaryotic initiation factor 2 (eIF2α) when UPR occured. “ (line 236- line 238); “In response to ER stress, a portion of the membrane-bound 90 kDa ATF6 undergoes proteolytic cleavage, releasing the soluble 50 kDa protein with an ability to induce transcription” (line 240- line 242); “Similar to PERK, inositol requiring 1 α(IRE1α) is one of ER transmembrane receptor during UPR, activated IRE1α splice mRNA of XBP1 through endoribonuclease and XBP1 translate to active protein eventually” (line 244- line 246); “We also investigate the role of C/EBP homologous protein (CHOP) which regulated ER stress associated apoptosis during severe impairment of ER functions” (line 248- line 249). “LC3 II and p62 are autophagy marker which LC3 II participate in membrane formation of autophagosome and p62 is possess to binding with ubiquitinated protein during autophagy. In addition, beclin-1 is one of component of class III PI(3)K complexes that involved in maturation of autophagosome and autophagy related protein 5 (ATG5) conjugated autophagy related protein 12-autophagy related protein 16 (ATG12-ATG16) complex promoted autophagasome formation” (line 396- line 402) to explain ATF6, eIF2α, XBP1s, CHOP, LC3 II, p62, beclin-1, ATG5 about their meaning and function respectively.

5.- As long as the authors do not correct these points from this formal point of view, I will not be able to approve the acceptance of the manuscript.

Ans: We correct these points which were suggested by the reviewer.

6.- The authors abuse the word "downstream" and it is not at all necessary in this context, therefore, it must be corrected.

 Ans: We rechecked and reduced it with only used the word “downstream” in sentence “OSGIN1 is a downstream gene of p53 which could elevate cellular ROS production”. We also rewrote the sentence as “Here, we investigated the roles of spliced form of X-box binding protein 1 (XBP1s) target gene oxidative stress induced growth inhibitor 1 (OSGIN1) in PA-induced vascular dysfunction. (line 19- line 21)”

Background aspects of the work

1.- It is very difficult to understand that the negative effects of a compound, in this case palmitic acid, on vascular lipotoxicity is also the solution. If PA causes increases in XBP1 levels, and this lipotoxicity is then characterised by increases in the IRE1-α/OSGIN1 axis, it cannot be said that this axis protects against lipotoxicity. In any case, it should be said that the blockage of this axis is responsible for the protection. And this should be made explicit both in the title and throughout the text.

Ans: We rewrote the sentences as “These results proved that XBP1s protect PA-induced endothelial dysfunction through eNOS expression by evidence of silence XBP1s. (line 350-352)”; “Collectively, these results indicate that XBP1s protect endothelial cells via autophagy under lipotoxic conditions through silencing of XBP1s. (line 412-line 414)”

2.- It is very strange that with only three experiments, on the one hand, such a large statistical range is obtained in most cases (the authors should show the individual values of these experiments in supplementary files), and on the other hand, that in some cases there are significant differences.

Ans: We checked all the individual value of these experiments and related figure, and upload these results in supplemental data as “Table S2: Individual value of related figure”. (line 762)

3.-An important example of this occurs in an important case for the work, i.e., variations in eNOS expression. The authors say that there are very significant differences when it does not seem so. In particular, figure 3D. It is very difficult to believe that with the expression data shown there could be significant differences between siXBP1+BSA versus siXBP1 +PA. Reanalyse similar ones.

Ans: We reanalysed the eNOS data, and the result was significant differences between siXBP1+BSA versus siXBP1+PA as p-value was 0.0221; After reanalysed the p-eNOS data, there was no significant differences between siXBP1+BSA versus siXBP1 +PA as p-value was 0.9175.

4.- Throughout the manuscript there are sentences that we do not understand why they are included, or at least the authors do not clarify or justify them adequately. For example, in Results 3.3 there is a sentence that is not adequately explained: "eNOS-mediated cell migration contributes to angiogenesis [29]".

Ans: We rewrote the sentence as “eNOS-mediated cell migration contributes to angiogenesis [29], herein western blot analysis was used to verify the role of eNOS in cell migration under silence of XBP1s during PA induced vascular lipotoxicity.” (line 345- line 349)

5.- The schematic figure should be modified because, if PA causes increases in ATF6, IRE1-α, elF-2α, XBP1a, OSGIN1 and LC3II, these markers cannot, at the same time, be the protectors of the lipotoxicity caused by PA itself.

Ans: We remake the schematic figure as below (line 726- line 741)

Round 2

Reviewer 2 Report

Comments to the authors

Most of the suggestions made in my first review have been corrected by the authors, although they or the journal itself should provide a version of the manuscript in a docx file with the accepted changes to see if the changes are really appropriate. However, I still have a big doubt that the authors do not finish explaining. According to what was proposed by the authors, the working hypothesis is the following: Palmitic acid causes an important endothelial dysfunction that is manifested by a significant increase in the levels of IRE-alpha in the endoplasmic reticulum and, as a consequence, the levels are increased. expression of the XBP1s/OSGIN1/LC3 II pathway. All this facilitates autophagy and finally cell migration. When XBP1s is silenced under these conditions, all these molecular and cellular events are reversed. Therefore, I believe that it is the blockade of OSGIN1, after the silencing of XBP1s, that protects from the harmful effects of palmitic acid and therefore the authors should correct this both in the title and in the rest of the manuscript. On the other hand, if I am wrong in the assessment, the authors must explain it clearly and exhaustively.

Round 3

Reviewer 2 Report

Although, the authors do not finish answering the crucial question of this work: Is it possible that the molecule that generates a lipotoxicity problem, palmitic acid, is also the one that solves this problem, and this is serious from my point of view, the rest of the suggestions have been solved.